# Association of Physical Performance, Muscle Strength and Body Composition with Self-Assessed Quality of Life in Hemodialyzed Patients: A Cross-Sectional Study

**DOI:** 10.3390/jcm11092283

**Published:** 2022-04-20

**Authors:** Maja Nowicka, Monika Górska, Krzysztof Edyko, Magdalena Szklarek-Kubicka, Adam Kazanek, Malwina Prylińska, Maciej Niewodniczy, Tomasz Kostka, Ilona Kurnatowska

**Affiliations:** 1Department of Internal Medicine and Transplant Nephrology, Medical University of Lodz, 90-153 Lodz, Poland; maja.a.nowicka@gmail.com (M.N.); monika.gorska@umed.lodz.pl (M.G.); krzysztof.edyko@gmail.com (K.E.); 2Dialysis Department, Norbert Barlicki Memorial Teaching Hospital No. 1, 90-153 Lodz, Poland; szklarekm@gmail.com; 3Therapeutic Rehabilitation Outpatient Clinic, Medical Center Lodz Baluty, 91-745 Lodz, Poland; ae.kaz@op.pl (A.K.); prylinskam@gmail.com (M.P.); 4Rehabilitation Department, Norbert Barlicki Memorial Teaching Hospital No. 1, 90-153 Lodz, Poland; maciekn@orange.pl; 5Department of Geriatrics, Healthy Ageing Research Center, Medical University of Lodz, 90-647 Lodz, Poland; tomasz.kostka@umed.lodz.pl

**Keywords:** chronic hemodialysis, muscle strength, quality of life, bioimpedance, nutrition, renal disease

## Abstract

(1) Patients on chronic hemodialysis (HD) experience impaired quality of life (QoL). We analyzed HD’s relationship with physical performance, body composition, and muscle strength; (2) QoL was assessed with the Short Form-36, composed of physical (PCS) and mental (MCS) health dimensions. Physical performance was assessed with the Short Physical Performance Battery (SPPB), body composition (lean tissue mass% (LTM%), fat tissue mass% (FTM%), and skeletal muscle mass% (SMM%)) was assessed with bioelectrical impedance, and lower extremity strength was assessed with a handheld dynamometer; and (3) we enrolled 76 patients (27 F, 49 M), age 62.26 ± 12.81 years, HD vintage 28.45 (8.65–77.49) months. Their QoL score was 53.57 (41.07–70.64); their PCS and MCS scores were 52.14 (38.69–65.95) and 63.39 (44.64–76.79) and strongly correlated (*p* < 0.0001, R = 0.738). QoL correlated positively with SPPB (R = 0.35, *p* ≤ 0.001), muscle strength (R from 0.21 to 0.41, *p* < 0.05), and LTM% (R = 0.38, *p* < 0.001) and negatively with FTM% (R = −0.32, *p* = 0.006). PCS correlated positively with SPPB (R = 0.42 *p* < 0.001), muscle strength (R 0.25–0.44, *p* < 0.05), and LTM% (R = 0.32, *p* = 0.006) and negatively with FTM% (R = −0.25, *p* = 0.031). MCS correlated positively with SPPB (R = 0.23, *p* = 0.047), SMM% (R = 0.25; *p* = 0.003), and LTM% (R = 0.39, *p* < 0.001) and negatively with FTM% (R = −0.34; *p* = 0.003). QoL was unrelated to sex (*p* = 0.213), age (*p* = 0.157), HD vintage (*p* = 0.156), and BMI (*p* = 0.202); (4) Better physical performance, leaner body composition, and higher muscle strength are associated with better mental and physical QoL in HD.

## 1. Introduction

Hemodialysis (HD) remains the most common form of kidney replacement therapy, yet the patients undergoing it present with significantly worse quality of life (QoL) compared with healthy individuals, kidney graft recipients, and those undergoing peritoneal dialysis [1,2,3,4].

Treatment length and frequency impose major changes in daily routine, domestic and social life. Most HD patients suffer from chronic fatigue, resulting from, i.a., chronic toxemia, anemia, disturbances in electrolyte levels, metabolic acidosis, and chronic inflammation [5]. Their condition imposes severe fluid and dietary restrictions; furthermore, HD patients often suffer from malnutrition, protein-energy wasting, and disease-induced sarcopenia [6], with an additional component of dynapenia—muscle weakness independent of its mass depletion. Dynapenia supposedly results from muscle fiber atrophy, disturbed muscle relaxation, and an increased amount of non-contractile tissue [7,8,9]. Moreover, end-stage kidney disease (ESKD) and chronic HD are associated with a shift in body composition; besides a decrease in lean tissue mass, some patients experience additional fat tissue loss, which is a poor outcome indicator, while others gain fat tissue, which is associated with decreasing QoL [10]. Immobilization associated with dialysis, high comorbidity burden, and overall lack of physical activity further decrease physical fitness and can lead to functional disability. These everyday impediments and limitations result in a high incidence of depression and anxiety in this population [11,12,13,14].

QoL forms an independent therapeutic target in ESKD, as low scores in QoL are linked to higher morbidity and mortality [15,16,17]. A multidimensional approach to QoL determinants in HD, including assessment of sarcopenia and dynapenia, has been undertaken in several studies. Methods of muscle strength evaluation usually rely on handgrip strength, rather than a more comprehensive assessment of several muscle groups; therefore, the aim of our study was to comprehensively investigate the relationships among objectively measured parameters: comorbidity burden, physical performance, muscle strength, body composition, and laboratory parameters with the physical and mental QoL of HD patients.

## 2. Materials and Methods

ESKD patients, aged 18–90 years, undergoing HD three times per week for at least 3 months from a single dialysis center were enrolled in the study.

The exclusion criteria included severe or acute cardiovascular, respiratory, musculoskeletal, neurological or liver conditions, i.e., active malignancies, acute infection, active bleeding, and severe anemia with hemoglobin level <8 mg/dL.

This study was approved by the local bioethical committee. Written informed consent was given by all participants.

### 2.1. Study Design

We collected data regarding demographics and anthropometrics, comorbidities, self-assessed quality of life, body composition, muscle strength, and physical performance. All measurements were performed by trained personnel before the mid-week dialysis session. All data were anonymized prior to statistical analysis.

#### 2.1.1. Demographic and Clinical Data

Data regarding patients’ age, sex, etiology of ESKD, dialysis vintage, and comorbidities were collected from patient interviews and medical records.

#### 2.1.2. Self-Assessed Quality of Life

Patients were asked to answer the SF-36 questionnaire, assessing self-reported functional health and wellbeing in a 4-week period. The participants completed it themselves andhad the opportunity to ask investigators for clarification. The questionnaire contains 35 questions, grouped into 8 domains, namely physical functioning (PF), role—physical (RP), bodily pain (BP), general health perceptions (GH), vitality (V), social functioning (SF), role—emotional (RE) and mental health (MH). These form two primary components: physical component summary (PCS = PF, RP, BP, and GH) and mental component summary (MCS = V, SF, RE, and MH). The 36th question—health change (HC)—assesses perceived change in health status over the previous year. Each answer is then coded on a scale from 0 to 100, with 100 corresponding to the best quality of life [18].

#### 2.1.3. Comorbidities

Coexisting conditions were assessed using the Charlson Comorbidity Index (CCI), modified by excluding patients’ age (utilized as an independent coefficient) and presence of moderate-to-severe kidney disease, which was applied in each case. CCI is a scale originally consisting of 19 diseases, the weight of which is based on the strength of their association with mortality [19].

#### 2.1.4. Physical Performance

We evaluated physical performance with the Short Physical Performance Battery (SPPB), a tool designed to assess lower extremity functioning in older patients, which reflects the activities of daily living. It estimates the risk of disability or its progression, institutionalization, hospitalization, and death. SPPB comprises 3 tests: gait speed, chair stand time, and balance. The total scores range from 0 to 12, with 0 being the worst performance [20].

#### 2.1.5. Muscle Strength

The patients were subjected to a muscle strength assessment. The maximal voluntary force of five muscle groups (quadriceps femoris, biceps femoris, iliopsoas with rectus femoris, triceps surae, and tibialis anterior) of both lower extremities was measured with a belt-stabilized handheld dynamometer (Microfet 2, Hogan Health Industries). All tests were performed according to the instruction manual; the testing positions are presented in Table 1. Each movement was tested three times, and the average scores were included in the analysis.

#### 2.1.6. Anthropometric Data and Body Composition

Height and body mass were measured before and after HD sessions; current dry body mass was collected from the patients’ medical HD records. Body mass index (BMI) was calculated for current dry body mass. Patients’ overhydration (OH) and body composition (lean tissue mass (LTM), mass percentage (LTM%), and index (LTI) and fat tissue mass (FTM), mass percentage (FTM%), and index (FTI)) were assessed using an electrical bioimpedance analyzer (BCM Fresenius™). Skeletal muscle mass (SMM), mass percentage (SMM%), and index (SMI) were calculated using bioimpedance data and equations developed by Janssen et al. [21].

#### 2.1.7. Laboratory Parameters

We retrospectively collected the following laboratory parameters assessed during routine monthly check-ups: for up to one month prior to the study visit, red blood cell parameters—RBC, HGB, MCV, and MCHC—and iron balance parameters—iron, ferritin, transferrin saturation, and total iron-binding capacity; from three months prior, calcium, phosphates, and parathormone.

### 2.2. Statistical Analysis

Continuous data were presented as means with standard deviations (SD) and were compared between the patient groups with an unpaired Student’s t-test. Ordinal data were compared with the Mann–Whitney U test. Nominal data were presented as numbers with percentages and were compared with Chi2 test or with Fisher’s exact test depending on the number of individuals in subgroups. Cronbach’s alpha was calculated to assess the internal consistency of SF-36 subcomponents in the hemodialyzed patients. Correlations between ordinal variables, such as patient performance on different scales, were tested with the Spearman rank correlation test. *p* values lower than 0.05 were considered statistically significant.

All analyses were performed with Microsoft Excel (Microsoft, Redmond, WA, USA) and with Statistica 13.0 software (StatSoft Polska, Kraków, Poland).

## 3. Results

### 3.1. Demographic and Clinical Data

Out of 164 patients undergoing HD in the center, 76 (27 F; 49 M) qualified for the study. The mean age was 62.26 years (SD 12.81), and the median time on HD was 28.45 (8.65–77.49) months. The etiology of ESKD was glomerulonephritis (26.3%), hypertensive (18.4%) and diabetic (18.4%) nephropathy, polycystic kidney disease (11.8%), other, (14.5%), or unknown (10.5%).

### 3.2. Self-Assessed Quality of Life

The median QoL score was 53.57 (41.07–70.64). The median scores for each domain and the health change score are presented in Table 2. Physical QoL (52.14, 38.69–65.95) and mental QoL (63.39, 44.64–76.79) had a strong positive correlation (*p* < 0.0001, R = 0.738).

The QoL score was unrelated to gender (*p* = 0.213), age (*p* = 0.157), or HD vintage (*p* = 0.156). On two instances, the scores of individual domains were significantly associated with these variables: women scored lower in the role—emotional domain (M = 83.33, 58.33–100.00; F = 66.67, 33.33–83.33; *p* = 0.026), while the physical functioning domain showed a weak inverse correlation with age (*p* = 0.0017, R = −0.355). Interestingly, the time on HD was inversely correlated with the health change scores (*p* = 0.010, R = −0.294).

### 3.3. Comorbidities

The patients’ comorbidities are presented in Table 3. The median score of CCI (excluding age and kidney disease) was 2 (1–3). The CCI index was positively correlated with age (R = 0.371, *p* < 0.001), and inversely with SPPB (R = −0.394, *p* < 0.001) and PCS (R = −0.235, *p* = 0.04) scores (Figure 1).

### 3.4. Physical Performance

The mean SPPB score was 8.3 (SD 3.15), which corresponds to mild limitations of physical function; exact scores are presented in Table 4. The SPPB scores inversely correlated with age (R = −0.373, *p* = 0.001) and were comparable between genders (*p* = 0.080).

SPPB scores correlated positively with QoL, PCS, MCS (Figure 1) and, interestingly, with both PF (R = 0.460, *p* < 0.001) and SF (R = 0.279, *p* = 0.017) as well as with RP (R = 0.255, *p* = 0.030).

### 3.5. Muscle Strength

The muscle strength measurements are presented in Table 5. Males presented with greater muscle strength across all muscle groups (*p* < 0.05). The strength of all muscle groups except the right triceps surae inversely correlated with the patient’s age; however, muscle strength did not correlate with HD vintage.

All muscle groups, except for the left and right tibialis anterior, showed weak to moderate positive correlations with QoL (Figure 2). All, except for the left tibialis anterior, correlated with the PCS; the strongest correlations were found with the biceps and quadriceps femoris. The strength of several muscle groups positively correlated with MCS as well. The strength of all muscle groups showed weak to moderate positive correlations with the SPPB score (R from 0.335 to 0.580; *p* < 0.05).

### 3.6. Anthropometric Data and Body Composition

The values of all body composition parameters are included in Table 6. Males had higher body mass, both pre-dialysis and dry body mass (*p* < 0.001 and *p* = 0.003, respectively), as well as a significantly larger skeletal muscle mass percentage (*p* < 0.001). They also presented with greater overhydration expressed as % of dry weight (OH%) than women (*p* = 0.029).

Patients’ age showed a weak, inverse correlation with SMI (R = −0.270, *p* = 0.019) and LTI (R = −0.254, *p* = 0.029); however, no significant relationship was observed with FTI (*p* = 0.357).

HD vintage was inversely correlated with BMI (R = −0.345, *p* = 0.002) and positively with OH% (R = 0.251, *p* = 0.032) but not with any other parameter of body composition.

Overall, QoL, PCS, and MCS scores correlated positively with lean tissue parameters but inversely with fatty tissue parameters (Figure 3). Percentage of lean tissue mass showed positive correlations with several SF-36 domains: physical functioning, role–physical, bodily pain, vitality, role—emotional, and mental health (R from 0.239 to 0.363, all *p* < 0.05); the percentage of muscle mass correlated with role—emotional and vitality (R = 0.293 and 0.235, *p* < 0.05). The percentage of fat tissue mass showed inverse correlations with not only role–physical but also with role—emotional, vitality, and mental health (R from −0.240 to −0.333; all *p* < 0.05). There was no significant correlation between QoL, PCS, or MCS and patients’ BMI, weight, and OH%.

### 3.7. Laboratory Data

SF-36 results, both the overall score and separate domains, were unrelated to the parameters of red blood cells, iron balance, and calcium-phosphate balance; laboratory data are presented in Table 7.

## 4. Discussion

The study evidenced a strong positive relationship between QoL and muscle strength, physical performance, a high percentage of lean tissue, and a low percentage of fat tissue in HD patients. While physical QoL showed the strongest correlation with muscle strength and physical performance, mental QoL showed the strongest correlation with body composition.

In our study, these correlations were stronger than those with more thoroughly studied factors such as sex, age, HD vintage, and comorbidity burden.

Contrary to most previous studies, we observed no association of overall QoL with sex or age [22,23]. However, with the components separated, older patients achieved significantly lower scores of physical QoL, while mental QoL scores remained independent of age, a pattern similar to that observed by Alshraifeen et al. [24]. We found no association between HD vintage and QoL; it is in agreement with Gerasimoula et al., yet contradicts the results of other studies. This discrepancy could be attributed to our relatively small study sample [23,25,26,27].

The comorbidity burden was assessed with the modified CCI scale, a predictor of one-year mortality and risk of disability and hospitalization [28,29,30]. In our study, patients with a higher comorbidity burden reported worse physical but not mental QoL, even though certain studies reported that high comorbidity is associated with poor mental health status in chronically ill populations [31,32,33].

We evaluated physical performance with the SPPB test, which Nogueira et al. proposed as a suitable screening method for decreased functional capacity in HD patients [34]. Oh et al. analyzed SPPB scores and QoL assessed with the EQ-5D questionnaire in an elderly population and found them to be correlated [35]. A large cohort study conducted on chronic kidney disease patients evidenced a graded association between severity of kidney dysfunction and worse SPPB scores; they also found SPPB scores correlating positively with physical QoL in this population [36]. Interestingly, we found that physical performance is related not only to physical QoL but also to mental QoL and social functioning. This is in concordance with Gómez et al., who reported poor SPPB scores to be associated with symptoms of depression and decreased cognitive functions [37].

One of the most noteworthy observations from our study was that a combination of low adiposity and high lean tissue content has a notably positive relationship with physical QoL and even more so with mental QoL. Likewise, Kalantar-Zadeh et al. found that greater tissue fat content was negatively associated with QoL scores of HD patients [10]. Our findings further support the idea of a bilateral link between obesity and poor mental health. A meta-analysis by Luppino et al. evidenced that individuals with obesity are more prone to developing depression, while individuals with depression are more prone to becoming obese [38]. Some suggest that fat tissue content, regardless of BMI, is responsible for the phenomenon [39]. Supporting this hypothesis, in our study, BMI itself showed no significant correlations with QoL. Moreover, a recently published study investigating the relationship between QoL and body composition in individuals without obesity showed an association between body composition, QoL, and depression symptoms. In females, higher overall adiposity and a predominantly abdominal distribution of adipose tissue were associated with lower QoL. In males, higher QoL was observed in individuals with larger LTM, including greater lower extremity musculature assessed via thigh circumference [40]. These results confirm the importance of body composition assessments in routine clinical practice, as opposed to BMI, which is of limited significance especially in HD patients.

In a longitudinal study by Martinson et al. investigating the relationship between musculature and QoL of 74 HD patients, MRI was used to assess body composition. The results indicate that patients with bigger mid-thigh muscle areas enjoyed higher scores in both mental and physical QoL [41]. Similarly, in our population, higher muscle mass was associated with better QoL. To better understand the relationship between musculature and QoL, we included another variable—muscle strength. Recent studies point towards the importance of dynapenia—a state of decreased muscle strength without the component of decreased muscle mass. Souweine et al. reported that patients who suffer from dynapenia are at a greater mortality risk despite being younger and having less comorbidity burden than those with sarcopenia [42]. In healthy adults, higher LBM is associated with greater upper body strength, while FTM is inversely correlated with lower extremity performance [43]. Interestingly, the correlations we found between QoL and muscle strength were more powerful than those with muscle mass. Of note, patients with greater muscle strength presented with higher mental QoL as well. This phenomenon has been described in the general population: a prospective study by Kandola et al. concluded that low hand-grip strength was a risk factor for depression and anxiety [44]. One of the hypotheses connecting muscular strength with mental QoL relates to brain-derived neurotrophic factor (BDNF). BDNF drives neurogenesis in the hippocampus and is produced in skeletal muscles. A decreased contraction of skeletal muscles can cause a decline in the secretion of BDNF as well as a volume reduction of the hippocampus and, thus, has been implicated in life satisfaction [45].

To the best of our knowledge, there are no studies comparing ESKD-related loss of function of different muscles or the association of different muscle groups’ strength to QoL in HD patients. Numerous chronic diseases, as well as aging (starting at approximately 50 years of age), accelerate the depletion of muscle mass and strength [46,47]. Johansson et al. revealed that the age-related decline in lower-body muscles and bigger muscles is more pronounced than that of smaller and upper-body muscles; therefore, an assessment of lower extremities muscle strength is more specific in sarcopenia detection than handgrip strength [48]. We hypothesized that the differences in disease-related muscle loss may be similar to that of an age-related phenomenon, as in our study, the strongest correlations between muscle strength and QoL were observed in relation to large thigh muscle groups. Therefore, future studies should explore potential differences in ESKD-related muscle loss and its relationship with QoL and their possible clinical significance; evaluating sarcopenia based on thigh muscle strength rather than handgrip strength should also be considered.

The results of our study may indirectly highlight the importance of interventions aimed at increasing the level of physical activity among HD patients [49]. Meta-analyses of exercise interventions in this population support this notion. They show that both interdialytic and at-home exercise regimens improve the mental and physical QoL of HD patients [50,51]. Ultimately, considering the protective relationship of lean body mass and higher muscle strength on QoL, interventions aimed to increase muscle strength together with increasing lean tissue content, ought to be encouraged.

In contrast with previous reports of anemia being linked to worse QoL [10], we observed no significant relationship between QoL scores and HGB, RBC, or iron balance parameters.

There were some limitations to our study. Firstly, the study was cross-sectional, without prospective analysis, which was interrupted by the SARS-CoV 2 pandemic, and we could not perform morbidity and mortality analyses. Secondly, the relatively wide distribution of age, HD vintage, and comorbidity burden among our study subjects could have biased results, due to the small overall sample size. Thus, we cannot exclude the possibility that we failed to observe correlations that could have otherwise been significant in a larger study population. Another limitation of our study is its susceptibility to selection bias or the healthy patient bias, seeing as we only included walking patients without severe physical disabilities.

## 5. Conclusions

HD patients’ quality of life, both physical and mental, is strongly related to their muscle strength, body composition, and physical performance. From a clinical standpoint, interventions aimed at improving muscle strength and at increasing lean tissue mass may improve the mental and physical QoL. Repetitive assessments of these parameters should be implemented in clinical practice.

## Figures and Tables

**Figure 1 jcm-11-02283-f001:**
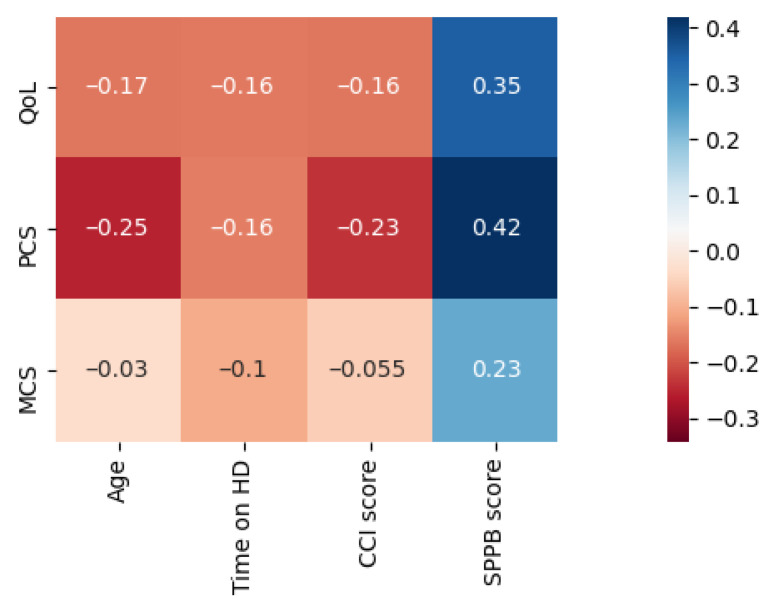
Correlation matrix visualizing correlations between SF-36 scores and age, HD vintage, CCI, and SPPB score. Positive correlations are visualized in shades of blue, and negative correlations are visualized in shades of red. Statistically significant correlations are marked with an asterisk.

**Figure 2 jcm-11-02283-f002:**
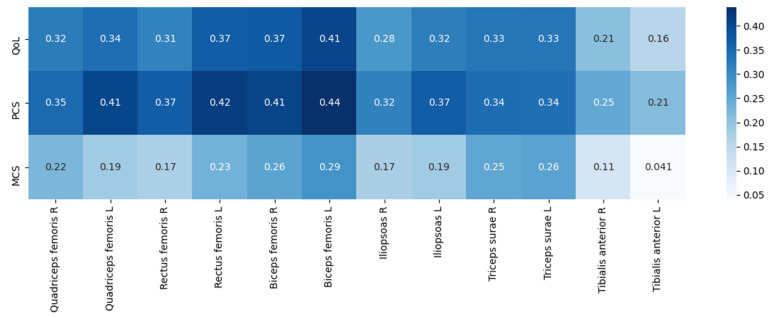
Correlation matrix visualizing correlations between SF-36 scores and muscle strength (Pearson’s test). Positive correlations are visualized in shades of blue, and negative correlations are visualized in shades of red. Statistically significant correlations are marked with an asterisk.

**Figure 3 jcm-11-02283-f003:**
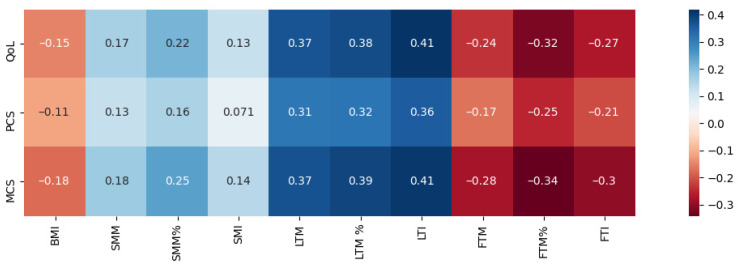
Correlation matrix visualizing correlations between SF-36 scores and body composition parameters (Spearman’s test). Positive correlations are visualized in shades of blue, and negative correlations are visualized in shades of red. Statistically significant correlations are marked with an asterisk.

**Table 1 jcm-11-02283-t001:** Muscle testing positions with transducer placement.

Muscle	Movement	Limb Position	Dynamometer Position
Quadriceps femoris	Knee extension	Sitting; hip and knee flexed 90 degrees	Just proximal to malleoli
Biceps femoris	Knee flexion	Sitting; hip and knee flexed 90 degrees	Just distal to malleoli over the Achilles tendon
Iliopsoas + rectus femoris	Hip flexion	Sitting; hip and knee flexed 90 degrees	Just proximal to femoral condyles
Triceps surae	Ankle plantarflexion	Supine; knee extended and ankle in neutral dorsiflexion	Over metacarpal phalangeal joints
Tibialis anterior	Ankle dorsiflexion	Supine; knee extended and ankle in neutral dorsiflexion	Just proximal to metacarpal phalangeal joints

**Table 2 jcm-11-02283-t002:** SF-36 scores.

	Score	Cronbach’s Alpha
Quality of Life (QoL)	53.57 (41.07–70.64)	0.945
Physical QoL (PCS)	52.14 (38.69–65.95)	0.908
Mental QoL (MCS)	63.39 (44.64–76.79)	0.92
Physical Functioning (PF)	57.50 (35.00–80.00)	0.908
Role—physical (RP)	50.00 (31.25–62.50)	0.86
Bodily Pain (BP)	55.00 (45.00–100.00)	0.875
General Health (GH)	40.00 (30.00–50.00)	0.483
Vitality (VT)	50.00 (34.36–62.50)	0.781
Social Functioning (SF)	62.50 (50.00–100.00)	0.865
Role—emotional (RE)	79.17 (50.00–100.00)	0.92
Mental Health (MH)	65.00 (50.00–80.00)	0.857
Health Change (HC)	6.58% major improvement	-
	13.16% minor improvement	
	27.63% no change	
	31.89% minor deterioration	
	19.74% major deterioration	

Scoring: 0—worst health, 100—best health. Nominal variables are presented as absolute and relative numbers and continuous data are presented as medians with the values of upper and lower quartiles.

**Table 3 jcm-11-02283-t003:** Charlson Comorbidity Index scores.

Comorbidities (CCI Weight in Points)	Number of Patients (%)
Myocardial Infarction (1)	19.7% (N = 15)
Congestive Heart Failure (1)	21.1% (N = 16)
Peripheral Vascular Disease (1)	17.1% (N = 13)
Cerebrovascular Disease (1)	13.2% (N = 10)
Dementia (1)	2.6% (N = 2)
Chronic Pulmonary Disease (1)	11.8% (N = 9)
Connective Tissue Disease (1)	9.2% (N = 7)
Peptic Ulcer Disease (1)	26.3% (N = 20)
Mild Liver Disease (1)	2.6% (N = 2)
Moderate or Severe Liver Disease (3)	1.3% (N = 1)
Diabetes Mellitus (1)	10.5% (N = 8)
Diabetes Mellitus with chronic complications (2)	22.4% (N = 17)
Hemiplegia (2)	5.3% (N = 4)
Leukemia (2)	-
Lymphoma (2)	-
Solid tumor (2)	9.2% (N = 7)
Metastatic Solid Tumor (6)	-
AIDS (6)	-
Median score [points]	2 (1–3)

**Table 4 jcm-11-02283-t004:** Short Physical Performance Battery scores.

Level of Disability	Percentage of Patients (N)
10–12 points—no limitations	43.42% (N = 33)
7–9 points—mild limitations	26.32% (N = 20)
4–6 points—moderate limitations	18.42% (N = 14)
0–2 points—severe limitations	5.26% (N = 4)
Mean score (points)	8.30, SD 3.15

Nominal variables are presented as absolute and relative numbers, and continuous data are presented as means with standard deviations.

**Table 5 jcm-11-02283-t005:** Lower extremities muscle strength.

Muscle/Muscle Group	Mean Strength (Kilogram Force)
Quadriceps femoris R	16.99, SD 6.19
Quadriceps femoris L	15.98, SD 5.48
Biceps femoris R	13.31, SD 4.93
Biceps femoris L	12.92, SD 4.84
Iliopsoas and rectus femoris R	17.54, SD 6.29
Iliopsoas and rectus femoris L	16.68, SD 5.84
Triceps surae R	13.10, SD 4.81
Triceps surae L	12.51, SD 4.29
Tibialis anterior R	11.77, SD 5.15
Tibialis anterior L	12.20, SD 5.33

Data are presented as means with standard deviations. R, right; L, left.

**Table 6 jcm-11-02283-t006:** Body composition parameters.

Bioimpedance Parameter	Mean/Median Value
Dry weight (kg)	77.83, SD 16.52
Weight (kg)	78.56, SD 15.99
Height (cm)	169.9, SD 9.33
BMI (kg/m2)	27.09, SD 4.31
OH (L)	0.6 (−0.4–2.1)
OH% (%)	0.78% (−0.52–2.59%)
FTM (kg)	28.96, SD 11.87
FTM% (%)	36.0%, SD 10.4%
FTI (kg/m2)	10.03, SD 3.99
LTM (kg)	35.90 (29.70–44.10)
LTM% (%)	46.2% (40.4–56.3%)
LTI (kg/m2)	12.40 (11.10–14.30)
SMM (kg)	25.79, SD 6.43
SMM% (%)	33.2%, SD 7.2%
SMI (kg/m2)	8.82, SD 1.56

Data are presented as means with standard deviations or as medians with the values of upper and lower quartile. BMI, body mass index; OH, overhydration; FTM, fat tissue mass; FTI, fat tissue index; LTM, lean tissue mass; LTI, lean tissue index; SMM, skeletal muscle mass; SMI, skeletal muscle index.

**Table 7 jcm-11-02283-t007:** Laboratory assessments.

Parameter	Mean/Median Value
RBC (10^12^/L)	3.55, SD 0.46
HGB (g/dL)	11.32, SD 1.17
MCV (fL)	94.44, SD 5.51
MCH (pg)	32.07, SD 2.08
Fe (mg/dL)	72.0 (56.0–91.0)
ferritin (mg/mL)	1039.0 (540.2–1533.0)
TS (%)	31.0% (24.0–44.0%)
TIBC (mg/dL)	228.4, SD 34.1
PTH (pg/mL)	245.0 (111.5–454.4)
Ca (mmol/L)	2.23, SD 0.21
phosphates (mmol/L)	1.93, SD 0.47

Data are presented as means with standard deviations or as medians with the values of upper and lower quartile. RBC, red blood cells; HGB, hemoglobin; MCV, mean corpuscular volume; MCH, mean corpuscular hemoglobin; Fe, iron; TS, transferrin saturation; TIBC, total iron-binding capacity; PTH, parathormone; Ca, calcium.

## Data Availability

The datasets used and/or analyzed during the current study are available from the corresponding author upon reasonable request.

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
