# Peer review of "Association of Physical Performance, Muscle Strength and Body Composition with Self-Assessed Quality of Life in Hemodialyzed Patients: A Cross-Sectional Study"

_jcm, 2022, doi:10.3390/jcm11092283_

Round 1
Reviewer 1 Report
In my opinion the paper needs further deepening on the discussion. As many of the items are not properly discussed.
Also a major issue is related to references as authors refer to a specific citation but the reference is dicerse. For example in line 280-282 or 287-289.
Also references must help the reader to deep in the theme but the references are pf studies not intended to explore a specific topic.
Other than that the design is adequate but a full revision must be considered prior to publication.
Author Response
Thank you for pointing the issue with the references out. The Reviewer is correct as our references were sorted alphabeticaly by mistake; we have adjusted and double-checked the references order.
As suggested by Reviewer the discussion has been significantly expanded, while taking into account other Reviewers recommendations (lines 283-288, 302-303, 309-326).
Reviewer 2 Report
Nowicka and colleague report in an article with the title „Association of physical performance, muscle strength and body composition with self-assessed quality of life in hemodialyzed patients: a cross-sectional study“ about chronic hemodialysis (HD) patients, who experience impaired quality of life (QoL). They analyzed its relationship with physical performance, body composition and muscle strength; QoL was assessed with Short Form-36, composed of physical (PCS) and mental (MCS) health dimensions. Physical performance was assessed with Short Physical Performance Battery (SPPB), body composition (lean tissue mass% LTM%, fat tissue mass% FTM%, skeletal muscle mass% SMM%) with bioelectrical impedance and lower extremity strength with handheld dynamometer. They enrolled 76 patients (27F, 49M), age 62.26±12.81 years, HD vintage 28.45 (8.65-77.49) months. QoL score was 53.57 (41.07-70.64); PCS and MCS scores were 52.14 (38.69-65.95) and 63.39 (44.64-76.79) and strongly correlated (p<0.0001, R=0.738). QoL correlated positively with SPPB (R=.35, P=<.001), muscle strength (R from .21 to .41, P<.05), LTM% (R=.38, P<.001), negatively with FTM% (R=-.32, P=.006). PCS correlated positively with SPPB (R=.42 P<.001), muscle strength (R.25-.44, P<.05), LTM% (R=.32, P=.006), negatively with FTM% (R=-.25, P=.031). MCS correlated positively with SPPB (R=.23, P=.047), SMM% (R=.25; P=.003), LTM% (R=.39, P<0.001), negatively with FTM% (R=-.34; P=.003). QoL was unrelated to sex (P=.213), age (P=.157), HD vintage (P=.156), BMI (P=.202); (4) Better physical performance, leaner body composition, higher muscle strength are associated with better mental and physical QoL in HD.
Major concerns: The strength of different muscles correlate different with the measurements for quality of life. Why is this? Why has the biceps femoris the best correlation?
Author Response
Thank you for bringing this to our attention.
ESKD, especially in chronic HD patients is associated with muscle atrophy. To the best of our knowledge, there are no studies comparing ESKD-related function loss of different muscles or the association of different muscle groups to QoL in HD patients. Numerous chronic diseases, as well as aging (starting at approximately 50 years of age), accelerate depletion of muscle mass and strength. Johansson et al. study revealed that the decline of age-related lower-body and bigger muscles is more pronounced than that of smaller and upper-body muscles; therefore, assessment of lower extremities muscle strength is more specific in sarcopenia detection than handgrip stregth. We hypothesize the differences in disease-related muscle loss may be similar to that of age-related phenomenon, as in our study strongest correlations between muscle strength and QoL were observed in relation to big thight muscle groups.
Therefore, future studies should explore potential differences in ESKD-related muscle loss and its relationship to QoL and their possible clinical significance; it should be also considered to evaluate sarcopenia based on thight muscle strength rather than handgrip strength.
As we feel this aspect of the study has not been sufficiently discussed before, we have described it more broadly in the Discussion section (lines 314-326).
Reviewer 3 Report
In this manuscript, the authors examine objective parameters of body composition with SF-36 scores and quality of life indicators. It is an interesting study of perception of QOL with substantive measurements in patients with ESKD.
Some findings are novel and worth attention such as the lack of association between HD vintage and QOL.
A few issues need to be addressed:
In the methods, the authors saythat they excluded patients with severe cardiac, pulmonary disease and malignancy; however inTable3 there are several patients with lymphoma, leukemia, solid tumor etc.. On the whole it is unclear why some of these would be excluded as co-morbidities are an integral part of ESKD. This data would be more relevant and generalizable if those were included rather than excluded.
Author Response
Thank you for bringing this issue to our attention.
In order to make sure that the detected relationships are significant especially in the context of changes associated with chronic HD, we decided to exclude from the study patients with diseases that could significantly affect not only their quality of life or physical fitness, but also the possibility of completing the study assessments (SPPB test, muscle strength measurements). We did not exclude patients with comorbidities typical of chronic kidney disease (such as vascular disease, heart disease, COPD, non-active malignancies) as shown in the CCI results, but only patients with its’ severe or acute course.
In our study population there were no patients with lymphoma, leukemia and metastatic solid tumors; the values presented in brackets refer to scores used to calculate CCI. We understand that the way the table has been formatted may be confusing, therefore we have corrected it (Table 3).
Round 2
Reviewer 1 Report
I see that changes have been performed and the manuscript improved. No contraindications for publication.